# Pharmaceutical Development and Design of Thermosensitive Liposomes Based on the QbD Approach

**DOI:** 10.3390/molecules27051536

**Published:** 2022-02-24

**Authors:** Dorina Gabriella Dobó, Zsófia Németh, Bence Sipos, Martin Cseh, Edina Pallagi, Dániel Berkesi, Gábor Kozma, Zoltán Kónya, Ildikó Csóka

**Affiliations:** 1Faculty of Pharmacy, Institute of Pharmaceutical Technology and Regulatory Affairs, University of Szeged, Eötvös u 6, H-6720 Szeged, Hungary; nemeth.zsofia@szte.hu (Z.N.); sipos.bence@szte.hu (B.S.); cseh.martin@szte.hu (M.C.); pallagi.edina@szte.hu (E.P.); csoka.ildiko@szte.hu (I.C.); 2Department of Applied and Environmental Chemistry, Faculty of Science and Informatics, Institute of Chemistry, University of Szeged, 1, Rerrich Béla tér, H-6720 Szeged, Hungary; daniel.berkesi@gmail.com (D.B.); kozmag@chem.u-szeged.hu (G.K.); konya@chem.u-szeged.hu (Z.K.)

**Keywords:** quality by design, quality planning, initial risk assessment, critical factors, thermosensitive liposomes, thin-film hydration method

## Abstract

This study aimed to produce thermosensitive liposomes (TSL) by applying the quality by design (QbD) concept. In this paper, our research group collected and studied the parameters that significantly impact the quality of the liposomal product. Thermosensitive liposomes are vesicles used as drug delivery systems that release the active pharmaceutical ingredient in a targeted way at ~40–42 °C, i.e., in local hyperthermia. This study aimed to manufacture thermosensitive liposomes with a diameter of approximately 100 nm. The first TSLs were made from DPPC (1,2-dipalmitoyl-sn-glycerol-3-phosphocholine) and DSPC (1,2-dioctadecanoyl-sn-glycero-3-phosphocholine) phospholipids. Studies showed that the application of different types and ratios of lipids influences the thermal properties of liposomes. In this research, we made thermosensitive liposomes using a PEGylated lipid besides the previously mentioned phospholipids with the thin-film hydration method.

## 1. Introduction

Liposomes are spherical formations, vesicles located by a membrane bilayer, in the nanometre size range [1]. The first liposomes were made by Bangham et al. in 1965 [2]. The inner phase of liposomes is separated from the dispersed media by a double lipid layer. The size of these vesicles can range from 20 nm to some μm. The thickness of the membrane is around 4 nm [1]. Phospholipids are important components of liposomes that, due to their hydrophobic tails and hydrophilic heads, are able to form a double membrane layer. One of their crucial pharmaceutical advantages is that these vesicles can incorporate hydrophobic and hydrophilic active pharmaceutical ingredients (APIs) as well. The hydrophilic drug is found in the hydrophilic core of the liposomes, while the lipophilic API is in the wall of the vesicles [1]. Liposomes may contain more than just a conventional drug. Recently, there has been an increase in the use of gene therapies, specifically in tumour therapy and the treatment of congenital genetic diseases [3,4,5,6,7]. Liposomes are also very useful in this area, as they can deliver genes to cells. In addition to the pharmaceutical industry, they are also widely researched and applied by the cosmetics and food industries. Liposomes can be produced in numerous ways. The thin-film hydration technique, also known as the Bangham method—named after Alec D. Bangham, the developer of the practice [2]—is one of the most frequently used preparation methods. This method is still widely used for the typical production of non-phospholipid/phospholipid vesicles located in a membrane bilayer and formed by supramolecular assembly [8,9,10,11]. Its first step is to solve the lipids of the membrane and the hydrophobic API in an organic solvent, which is later removed. This elimination can happen via lyophilisation or vacuum evaporation. The hydration of the film follows this step. The most often used hydration media are aqueous solutions: buffers, physiological saline solution, or solutions of carbohydrates, etc. This hydration step always has to be performed above the phase transition temperature (Tm) of the lipids [12]. Liposomes are used broadly; their application in tumour therapy started at the beginning of the 1990s [13]. Numerous types of liposomes are known. One of them is grouped according to the properties of the ligands bound to their surface by the liposomes. Examples are conventional, PEGylated, immunoliposomes and bioresponsive liposomes. Bioresponsive liposomes include thermosensitive liposomes. Thermosensitive liposomes (TSL) contain phospholipids with very different Tm values. The membrane of the liposome transforms from its solid, gel-like form to a highly permeable form at Tm, whose state is typical for hydrophilic materials [14,15]. A more effective way to target the active ingredient of liposomes is to attach signalling and targeting molecules to the liposome (antibodies, peptides, oligosaccharides, etc.). They bind to the recognised specific target molecule located either on the cancer cells themselves or on the surface of the endothelium of the altered vascular system that encloses the tumour. The only drawback is the heterogeneity of the tumours. A variety of antigens can be expressed by a particular tumour, and it is not certain that a particular antibody will target what would be required in a given therapy. Thermosensitive liposomes release their active components in the appropriate area under the influence of energy transfer or a biological signal. Examples of such signals include temperature change, pH change, light exposure, and ultrasound [14]. Thermosensitive liposomes are vesicles used for drug delivery that release their API in a targeted way at ~40–42 °C, i.e., in local hyperthermia [14]. TSLs can be further grouped as low-temperature-sensitive liposomes (LTSL) and high-temperature-sensitive liposomes (HTSL). LTSLs release the incorporated API above the physiological body temperature (37 °C), while in the case of the HTSLs, it happens at even higher temperatures. The API release from LTSLs is induced by the hyperthermic area formed due to the extra heat originating from the increased metabolic processes of the tumour tissue. In the case of HTSLs, this release happens at a place which is heated by an external heat source. The energy arising from the external source may increase the temperature even higher than the tumour’s original value [16]. TSLs are more selective and stable than traditional (natural phospholipid-containing) liposomes. With the help of TSLs, 20–30 times greater API concentrations can be achieved in the targeted tissues than with the free drugs [17], and 5–10 times greater than with traditional liposomes [18]. Thus, for various diseases, local therapy can be performed (i.e., tumour therapy), which can decrease the toxicity of the procedure in parallel. It is favourable for the patient if the therapy is targeted because, in this way, the applied API does not damage the healthy areas. The first TSLs were made from DPPC (1,2-dipalmitoyl-sn-glycero-3-phosphocholine) and DSPC (1,2-dioctadecanoyl-sn-glycero-3-phosphocholine) phospholipids (DPPC:DSPC = 7:3) [19]. Studies showed that the application of different types and ratios of lipids influences the thermal properties of liposomes [14]. Later research demonstrated that the addition of PEGylated phospholipids to the lipid mixture influences the size [20] and the thermosensitivity of the liposomes [12,13].

Since the beginning of the 2000s, the quality by design (QbD) method has been taking the place of the ‘in-process study’-based process and quality control system (quality by testing = QbT) in the pharmaceutical industry. The QbD concept is a knowledge- and risk- assessment-based quality management approach, applied principally in the industry [21,22,23,24], but it can also be extended and applied in the early pharmaceutical research and development (R&D) phase [24]. The essential elements in a QbD approach are the following: determining the quality target product profile (QTPP), selecting the critical quality attributes (CQAs) and critical process parameters (CPPs), performing the risk assessment (RA), design of experiments (DoE), developing a design space (DS) with a proper control strategy, and finally managing the product lifecycle based on aspects of continuous improvement. These steps and definitions are presented in the relevant documents of the International Council for Harmonisation of Technical Requirements for Pharmaceuticals for Human Use (ICH) [17,18,19,22]. According to the ICH descriptions, CQAs are related to the quality, safety, and efficacy profile of the product. In contrast, critical material attributes (CMAs) and CPPs are connected to the selected production method. By performing RA using a risk estimation matrix, we can obtain the ranking of CQAs, CMAs, and CPPs by the degree of their impact on the targeted product quality. The factors that have the highest critical impact on the final product have to be the focus of the development process. These should be the key elements in the factorial design of the experimental work. The QbD-guided procedure in development and industrial manufacturing provides more information and knowledge about the final product and the manufacturing process and has advantages in marketing authorisation procedures [22,23]. Risk factors regarding the development of liposomes have already been analysed from different viewpoints by researchers worldwide. A general overview of the QbD approach for liposome formulation without a defined preparation method completed with characterisation techniques was provided by Porfire et al. [25]. Xu et al. analysed the size, encapsulation efficiency, and stability-affecting factors regarding the formulation, process, analytical method, and instrumentation reliability for liposomes prepared using the thin-film hydration method [26]. Although examples for quality analysis can be found in the literature [27], there have been no cases concerning application of the R&D QbD model on TSLs. Due to the emerging need for novel drug delivery systems capable of responding to the variety of pH, ionic, enzymatic, and thermal changes in the human body, quality management is a highly recommended factor in the development process [28,29,30]. Thus, the novelty of this work is the extension of the QbD approach to thermosensitive liposomes based on the previous statements of our research group in a risk-assessment-based development proposal for liposomes [31].

Our research team has been working on lipid-based nanoparticles for years. The aim of this study was to produce stable TSL liposomes with an average diameter of 100–200 nm using a QbD-based experimental design. Concerning the QbD assay, quality management tools such as editing an Ishikawa diagram and performing a risk assessment process were applied.

During the work, various wall-forming components (DPPC, DSPC, and DSPE-PEG3000 (1,2-distearoyl-sn-glycero-3-phosphoethanolamine-*N*-[amino(polyethylene glycol)-3000]), solvents (ethanol, methanol, and chloroform), and hydration media (phosphate-buffered saline solutions of different pHs and isotonic saline solution) were used to influence thermosensitive properties and particle size, characterised by material analysis methods using differential scanning calorimetry (DSC), thermogravimetry (TGA), dynamic light scattering (DLS), transmission electron microscopy (TEM), Fourier-transform infrared spectroscopy (FT-IR), and atomic force microscopy (AFM) techniques.

## 2. Materials and Methods

### 2.1. Materials

Two different compositions and two hydration media were used in this study to form thermosensitive liposomes.

The following phospholipids were used: DPPC-1,2-dipalmitoyl-sn-glycero-3-phosphocholine (Avanti Polar Lipids, Alabaster, AL, USA), DSPC-1,2-dioctadecanoyl-sn-glycero-3-phosphocholine (Avanti Polar Lipids, Alabaster, AL, USA), and DSPE-PEG3000-1,2-distearoyl-sn-glycero-3-phosphoethanolamine-*N*-[methoxy(polyethylene glycol)-3000] (Avanti Polar Lipids, Alabaster, AL, USA), solved in ethanol 96% (Molar Chemicals Kft., Budapest, Hungary). The excipients were used in different ratios (Table 1).

Phosphate buffer solution pH 7.4 (PBS pH 7.4) and sodium chloride physiological solution (saline solution) pH 5.5 [32] were used as hydration media. The composition of these solutions was as follows: PBS pH 7.4: 8.0 g/L NaCl, 0.20 g/L KCl, 1.44 g/L Na_2_HPO_4_ × 2 H_2_O, 0.12 g/L KH_2_PO_4_; saline solution: 0.9 g/L NaCl dissolved in distilled water. The materials used to make these hydration media were the following: sodium chloride (NaCl) (Molar Chemicals Ltd., Budapest, Hungary), potassium chloride (KCl) (Molar Chemicals Ltd., Budapest, Hungary), disodium hydrogen phosphate dihydrate (Na_2_HPO_4_ × 2 H_2_O) (Spektrum-3D Ltd., Debrecen, Hungary), and dipotassium phosphate (K_2_HPO_4_) (Spektrum-3D Ltd., Debrecen, Hungary).

None of the formulations contained active pharmaceutical ingredients (API).

### 2.2. Methods

#### 2.2.1. Elements of the QbD Design

##### Development of the Knowledge Space and Determination of the QTPP, CQAs, CMAs, and CPPs

In order to perform the initial RA, the first step was the collection of all the relevant influencing factors of the desired liposomal formulation. This collection and systemic evaluation are parts of the knowledge space development [33]. To evaluate the cause and effect relationships between the factors, an Ishikawa diagram [34] was set up, which helps determine the QTPP elements and identify the critical factors. QTPP is related to the quality, safety, and efficacy of the product, considering, e.g., the route of administration, dosage form, bioavailability, strength, stability, etc. [33,35]. In this study, the following was chosen as the QTPP: a nanosized, stable, thermo-responsive (thermosensitive) liposomal formulation for nasal administration capable of reaching the CNS.

CQAs are the physical, chemical, biological, or microbiological properties or characteristics that should be within an appropriate limit, range, or distribution to ensure the desired product quality, derived from the QTPPs and/or prior knowledge, and always dependent upon predefined goals. Other critical factors can be linked to the materials used, these are the CMAs, and the process factors whose variability has an impact on the quality of the final product are the CPPs [35]. In this experiment, the following CQAs were identified: the zeta potential, particle size, and morphology of the liposomes and their phase transition temperature (Tm). The group of the CMAs/CPPs in this study includes the preparation of the mixture, dissolution of the lipids, hydration medium, and the method of the stabilisation (drying/lyophilisation) process.

#### 2.2.2. Risk Assessment

The RA procedure was performed using the LeanQbD^®^ software (QbD Works LLC, Fremont, CA, USA, www.qbdworks.com accessed on 18 January 2022). The first element of this procedure was the interdependence rating between the QTPPs and the CQAs, followed by the same procedure between the CQAs and the CMAs/CPPs. A three-level scale was used to describe the relationship between the parameters: ‘high’ (H), ‘medium’ (M), or ‘low’ (L). Then, a risk occurrence rating of the CMAs/CPPs (or probability rating step) was made, and the same three-grade scale (H/M/L) was applied for the analysis. As the output of the initial RA evaluation, Pareto diagrams [36] were generated by the software, which presented the numeric data and the ranking of the CQAs and the CMAs/CPPs according to their potential impact on the desired final product (QTPP). The Pareto charts not only show the differences of the CQAs, CMAs, and CPPs by their effect but they also help to select the factors of a potential experimental design. A relative occurrence–relative severity chart was set up to visualise the severity of the CPP/CMA elements compared to each other.

### 2.3. Design of Experiments (DoE)

The DoE was made according to the results of the initial RA. A “two-level 3-factor full factorial design” model was applied for the design of the experimental studies. 2^3^ = 8 experiments were performed with this model to determine the relationship between the selected factors and the quality of the final liposomal product. The factors applied in the factorial design were those which showed the highest critical effect on the desired product according to the RA. These are: the ratio of the phospholipids building up the liposomes: DPPC:DSPC:DSPE-PEG3000 (+1 value: 80:15:5 n/n%; −1 value: 70:25:5 n/n%); the type of the hydration medium (+1 value: physiological saline solution; −1 value: pH = 7.4 phosphate-buffered saline); and the method of physical stabilisation (+1 value: lyophilisation; −1 value: vacuum drying). The effect of these factors was measured by testing the character of the CQAs (zeta potential, particle size, morphology of the liposomes, and Tm) as a response.

### 2.4. Preparation of Thermosensitive Liposomes

The liposomes were prepared from DPPC, DSPC, and DSPE-PEG3000 phospholipids with the thin-film hydration method. As the first step, 2.5 mg/mL stock solutions were prepared from the lipids with ethanol (96% EtOH-Molar Chemicals). The prepared stock solutions were mixed according to the defined mole ratios of the lipids (Table 1). After that, the solvent was removed via vacuum evaporation (Büchi Rotavapor: 60 °C, 25 rpm) of the ethanol. This process resulted in the formation of a lipid film layer on the wall of the round-bottom flask. During the hydration step, the appropriate medium (pH = 7.4 PBS (phosphate-buffered saline) (self-prepared) or 0.9% NaCl solution-physiological saline (self-prepared)) was poured on the top of the film; then the flask was placed into a thermostated ultrasonic bath (60 °C, 30 min) to form the liposomes. To improve the size and size distribution values of the liposomes, the ultrasonic treatment was followed by membrane filtration. The shaping of the liposomes was performed in two steps via vacuum membrane filtration using a 0.45 µm (nylon membrane disk filter 47 mm, Labsystem Ltd., Budapest, Hungary), then a 0.22 µm membrane filter (Ultipor^®^ N66 nylon 6.6 membrane disk filter 47 mm, Pall Corporation, New York, NY, USA), while the vacuum was created by a vacuum pump (Rocker 400 oil-free vacuum pump, Rocker Scientific Co., Ltd. New Taipei City, Taiwan). The prepared liposome samples were immediately investigated for vesicle size, polydispersity, and zeta potential and then lyophilised for stability purposes. The samples were conserved by lyophilisation with a SanVac CoolSafe freeze dryer (LaboGeneTM Lillerod, Lillerød, Denmark). First, 1 or 2 mL of the samples was frozen at normal atmospheric pressure, gradually decreasing the temperature from +25 °C to −40 °C. The vacuum was created when the temperature of the samples reached the desired value, reducing the pressure to 0.01 atmosphere, where the samples were stored for 8–10 h. After this period, the temperature of the tray was increased manually, step by step, from −40 °C to +30 °C until the pressure reached the normal atmosphere.

The applied lipid compositions (mole ration of lipids) and hydration media are shown in Table 1.

### 2.5. Characterisation of the Liposomes

#### 2.5.1. Vesicle Size and Zeta Potential of the Liposomes

The dynamic light scattering (DLS) technique, which describes the size distribution of the vesicles, was used to determine the size and the polydispersity index of the liposomes. For a measurement, 1 mL of liposome suspension was used and then diluted. Zeta potential is the potential difference between the dispersion medium and the liquid layer adsorbed on the surface of the particles, which can be used—among other things—to investigate the stability of a suspension. Low absolute zeta potential values predict the aggregation of the dispersed particles, while high values show major repulsion between the particles. Suspensions are considered stable in the latter case. The measurements were taken three times for each sample. Our measurements were performed with a Malvern Zetasizer Nano ZS device.

#### 2.5.2. Differential Scanning Calorimetry (DSC) and Thermogravimetric Analysis (TGA) Investigations

During the thermogravimetric measurements, the samples were heated to the defined temperature in the given atmosphere and investigated for mass changes. Our measurements were carried out in an oxygen atmosphere in the temperature range of 25–70 °C with a Mettler Toledo TGA/DSC1 STARe System apparatus at a heating rate of 10 °C/min. The measurements were carried out on ~5–10 mg of the samples in all cases.

To determine the thermosensitivity of the liposomes, their size was checked, the samples were kept above the expectable phase transition temperature for a while, and then cooled back to room temperature and tested for size changes. There is a further, simpler, and faster method to determine the thermosensitive characteristic—differential scanning calorimetry (DSC) measurement. These investigations were performed in the temperature range of 25–70 °C on ~5 mg of the samples at a heating rate of 10 °C/min. The measurements were performed three times for each sample.

#### 2.5.3. Transmission Electron Microscopy (TEM) Measurements

The size, structure, and morphology of the liposomes were characterised by transmission electron microscopy (TEM) measurements. The TEM images were made with an FEI Tecnai G2 X-Twin HRTEM microscope (FEI, Hillsboro, OR, USA) with an accelerating voltage of 200 kV. Suspensions were prepared from the formulations with ethanol and then spread onto a copper grid coated with a 3 mm diameter carbon film. For particle size and distribution analysis, public domain image analyser software—ImageJ—was used (https://imagej.nih.gov/ij/index.html accessed on 20 January 2022).

#### 2.5.4. Fourier-Transform Infrared (FT-IR) Spectroscopy Measurements

Mid-infrared (MIR) spectroscopy provides information about the chemical bonds of the materials and, in the case of crystalline compounds, the rearrangements in the crystal structures. The results of functionalisation made on the liposomes were investigated with an Avatar 330 FT-IR Thermo Nicolet spectrometer equipped with an infrared light source and optics in absorbance mode. The measurements were made from powder samples in the 3500–400 cm^−1^ wavelength range with a spectral resolution of 4 cm^−1^. The samples were mixed and pulverised with KBr and then pressed to form pellets. Pure KBr pellets were used as references. The measurements were performed three times for each sample.

#### 2.5.5. Atomic Force Microscopy (AFM)

One drop of solution was pipetted onto freshly cleaved mica (Muscovite mica, V-1 quality, Electron Microscopy Sciences, Washington, DC, USA) for the experiment. The AFM images were obtained using the tapping mode on an NT-MDT SolverPro Scanning Probe Microscope (NT-MDT, Spectrum Instruments, Moscow, Russia) under ambient conditions. AFM tips type PPP-NCHAuD-10 manufactured by NANOSENSORS (Neuchâtel, Switzerland) was applied with a nominal radius of curvature of 2 nm and 15 μm length. The non-contact silicon cantilevers had a typical force constant of 42 N/m and a resonance frequency of 330 kHz. Further information on the tip—thickness: 4.0 μm, length: 125 μm, width: 30 μm.

### 2.6. Statistical Analysis

Data analysis, statistics, and graphs were made from the experimental data with the Microsoft^®^ Excel^®^ (Microsoft Office Professional Plus 2013, Microsoft Excel 15.0.5023.100, Microsoft Corporation, Washington, WA, USA), OriginPro^®^ 8.6 software (OriginLab^®^ Corporation, Northampton, MA, USA), and the JMP^®^ 13 Software (SAS Institute, Cary, NC, USA). One-way ANOVA with post-hoc Tukey test was applied using GraphPad Prism (GraphPad Software Inc., San Diego, CA, USA). *p* < 0.05 was considered statistically significant, comparing the size, PDI, and zeta potential of the formulations.

## 3. Results and Discussion

### 3.1. Initial Knowledge Space Development

According to the QbD methodology, the first step of this work was to systematically evaluate the literature and the accurate collection of all the relevant information. Nanosystems, such as liposomes, also require special attention for the formulation to deliver the expected product quality. The properties of the starting materials—each step of the manufacturing process, possible investigations, desired product quality, and possible therapeutic uses—must also be thoroughly examined. The development of the knowledge space was visualised by interpreting the classic 4M Ishikawa diagram (Figure 1).

The Ishikawa diagram later contributed to the risk assessment process and identified the range of possible QTPP, CQA, CMA, and CPP elements. Based on the previous work of our research group, our goal was to develop a carrier system that is suitable for human use due to its favourable properties with increased impact. The selected CQAs can be seen in Table 2.

Based on the defined QTPPs, our goal was to produce thermosensitive liposomes with a monodisperse size distribution around 100 nm, which can serve as API-loadable nanocarriers, provide higher drug release due to the thermal change, and are easily transported across biological membranes. The product must be physically stable, which is a critical parameter for both particle size and shelf-life. The possibility of targeting via alternative administration routes must always be considered as these routes allow for improved pharmacokinetics and bioavailability.

### 3.2. Risk Assessment

The QbD-based risk assessment consists of two steps: first, an interdependence rating must be established between the QTPP-CQA and the CMA/CPP-CQA elements. Figure 2 shows the three-level interdependence rating between QTPPs and CQAs, where relationships were characterised as high, medium, and low risk.

The CQAs chosen appear to be at high risk in at least one relation with the QTPP elements. A definite requirement for liquid colloidal dosage forms is the nanosize range as well as its distribution. Zeta potential has been assigned critical value because, under appropriate surface charge conditions, nanoparticles with a given morphology repel each other and keep the solution in a colloidal dispersed state. The phase transition temperature plays a major role in the site of the activity of the biological response, which determines through which administrative routes the system can release the drug from the potential drug-loaded liposome in an effective concentration, thereby achieving an enhanced therapeutic effect. The relationships between the CMA/CPP and CQA elements are shown in Figure 3.

The film hydration method is a complex, multi-step sequence of operations that involves risks not only in its entirety, but also in its sub-steps. The CMA/CPP elements always include the composition, which in this case is the amount of lipids that make up the membrane. In the process, several solvents are used, such as dissolution medium, effluent solvent, hydration medium, and solvent for dissolving the final product. Of course, the material properties of these fluids also influence the manufacturing sub-processes, which are directly related to the product characteristics. For freeze-drying as a drying method that increases physical stability, the previous method of our research group was used, which—although it can represent many critical parameters alone—is excellent for increasing the shelf-life of the formulations. Vacuum drying and filtration through the membrane determine the shape in addition to the particle size. However, since a fixed, nanosized pore size membrane was used, deformation is adequately successful with its application. The probability rating—i.e., the quantification of the risk impacts—is shown in Figure 4 in Pareto diagrams. The results obtained were calculated by the software, which presented the numeric data and the ranking of the CQAs and the CMAs/CPPs according to their potential impact on the desired final product (QTPP).

Based on the probability rating, the three membrane-forming lipids were assigned the highest severity score. Besides hydration time, the cryoprotectant also showed a high risk, so fixed concentration and time were chosen to optimise the formulation around these parameters. As for CQAs, no significant severity score difference was observed between particle size, zeta potential, phase transition temperature, and particle size distribution. As we chose to investigate only five CQAs, it is not a problem as the investigation methods allow the exact determination of these parameters. The relative occurrence–relative severity graph in Figure 5 supports the severity scores and gives a more visual representation of the influencing CMA/CPP factors.

### 3.3. Design of Experiments (DoE)

These results of the RA gave the basis of setting up a factorial experimental design. The selected factors came from the RA (the ratio of the phospholipids, the type of the hydration media, and the physical stabilisation method) and were applied on two levels. The pattern of the experiments and the factors are presented schematically in Figure 5. The variables are the CMAs, and the CPPs found to have the highest critical effect on the desired product mentioned above, and each of them was used on a minimum and a maximum level, as shown in Figure 6. The characteristics of the prepared liposomes (zeta potential, particle size, morphology of the liposomes, and Tm) were analysed (as a response) after each experiment.

### 3.4. Results of the Vesicle Size and the Zeta Potential Analysis

The samples were prepared after the QbD-based planning. The following table (Table 3) presents the size, zeta potentials, and PDI values of the liposomes measured after synthesis and two weeks later (in the meantime, the samples were stored at 4 °C) after heat treatment at 46 °C.

The mean vesicle size of the 80:15:5_PSS sample was 75 nm, and its zeta potential value was −2.74 mV; the mean vesicle size of the 80:15:5_PBS sample was 130 nm, and its zeta potential value was −3.46 mV; the mean vesicle size of the 70:25:5_PSS sample was 154 nm, and its zeta potential value was −2.98 mV; the mean vesicle size of the 70:25:5_PBS sample was 165 nm, and its zeta potential was −3.56 mV. According to the literature, the best circulation time can be obtained for vesicles with a size of 100 nm; thus, we will try to optimise the production process in our future experiments [37]. The measured zeta potential values met expectations; the applied lipids had no charges, so high values were not expected. The results were strengthened by one-way ANOVA with post-hoc Tukey HSD statistical analysis. After synthesis, the 80:15:5_PSS sample showed more significant size reduction and smaller PdI values compared to the other three formulations (80:15:5_PSS vs. 80:15:5_PBS/70:25:5_PSS/70:25:5_PBS **, *p* < 0.01). Concerning the zeta potential value, there was no significance found between the samples.

It is essential to note the correlation between the size values measured after synthesis when investigating the effect of different hydration media on the same compositions. Applying PSS, the size of the liposomes was smaller than that obtained with the same compositions when PBS was used. The size of the vesicles in the case of the samples made from the 80:15:5 lipid composition was: PSS (75 ± 2 nm) < PBS pH 7.4 (154 ± 5 nm), and the same tendency could be observed for the samples made from the 70:25:5 lipid compositions as well. However, the zeta potential values were slightly more negative when PBS solution was used. The ionic strength of the hydration medium influences the value of the zeta potential; the higher the ionic strength, the more compact the ion layer formed around the vesicles and, due to this fact, the higher the zeta potential [38]. In the presented case, the ionic strength of the hydration medium was: saline solution (0.15 M) < PBS pH 7.4 (0.16 M).

The samples were investigated in a dispersed state after two weeks of storage in a refrigerator (~4 °C); furthermore, these samples were kept above their phase transition temperature (46 °C) for 30 min as DLS measurements were rerun. 46 °C was chosen as the temperature of the treatment because the liposomes disintegrate above 40 °C. The change in the size of the liposomes proves their instability in a dispersed state that highlights the importance of stabilisation by lyophilisation. Based on the PDI values, the 80:15:5_PSS samples can be considered homogeneous after synthesis. After two weeks, all the samples were homogeneous except for the 70:25:5_PBS sample; however, after heat treatment, these data are not relevant. It can be assumed that the stability of the liposomes decreases without treatment because their size was reduced; however, this size reduction resulted in a homogeneous size distribution based on the PDI values. In the case of the 80:15:5 samples, a notable size increase was observed after heat treatment. The 70:25:5 samples hydrated with PBS behaved differently because there was a decrease in their size instead of growth, but the change in the size of the vesicles was recognisable. Based on the obtained results, it was concluded that the liposomes destabilise when kept above their phase transition temperature, which may later lead to API release. 80:15:5_PSS was significantly smaller than the others (**, *p* < 0.01). Regarding PDI of the samples, 80:15:5_PSS vs. 80:15:5_PBS showed no significance after storage, whilst 80:15:5_PSS vs. 70:25:5_PSS/70:25:5_PBS showed significant difference (**, *p* < 0.01). Although the PDI of 70:25:5_PSS was smaller, its size was larger, so the 80:15:5_PSS formulation still proved to be favourable even after two weeks of storage. The zeta potential value of 80:15:5_PSS also showed significant difference compared to 80:15:5_PBS and 70:15:5_PBS formulations (**, *p* < 0.01) but insignificance can be found vs. 70:25:5_PSS. The results of the DLS measurements led to the conclusion that the 80:15:5_PSS sample shows the most homogenous size distribution measured after synthesis, and two weeks later, thus this sample proved to be the most stable compared to the others. After heat treatment, concerning size, all formulations showed significant differences compared to each other (**, *p* < 0.01) whilst no significance was observed for the zeta potential values. The PDI values were significantly higher (80:15:5_PSS vs. 80:15:5_PBS **, *p* < 0.01) or lower (80:15:5_PSS vs. 70:25:5_PSS, ** *p* < 0.01) but were insignificant compared to 70:25:5_PBS.

DLS and zeta potential measurements were repeated after the synthesis of the 80:15:5_PPS sample (Table 4), before lyophilisation, but now at different temperatures. These temperatures were 10, 20, 30, 35, 37, and 40 °C.

Based on the measurement results, it can be stated that the size of liposomes is almost constant at the storage temperature (10–20 °C) and then increases continuously with increasing temperature. The results conclude that liposomes are destabilised by keeping them above their phase transition temperature, which will later lead to the release of their active ingredient.

#### 3.4.1. Results of the Differential Scanning Calorimetry (DSC) and the Thermogravimetric Analysis (TGA) Investigations

Figure 7 demonstrates the results of the TGA and DSC measurements for the 80:15:5_PSS sample; however, the same curve was obtained in every case. Based on the TGA curve, there was no significant change in the mass of the sample. The slight increase in weight can be considered a measurement error because, due to its sensitivity, the instrument can show a weight increase when such a small amount of sample is measured.

Figure 7 demonstrates the TGA and DSC results for the 80:15:5_PSS. Similar curves were obtained for all samples. Based on the TGA curve, there was no significant change in the mass of the sample. The slight increase in weight is a measurement error due to the small sample amount.

The stability of the freeze-dried samples can be characterized by the glass transition temperature (T_g_) [39]. Glass transition and melting temperatures of 26 °C and 57 °C were found in the DSC curve in Figure 7, respectively. Based on the literature references, DPPC melting transition is around 38 °C [39,40], however, it is well known that T_m_ can increase in lipid mixtures [41,42,43]. The thermodynamic parameters (ΔH_m_ and T_m_) provide important information on liposome stability and the pharmacokinetics of the API. The higher the ΔH_m_, the stiffer the bilayers are [39]. In all of our samples an enthalpy change (ΔH_m_) of 7.8 J/g were found.

#### 3.4.2. Results of the Fourier-Transform Infrared (FT-IR) Spectroscopy Measurements

The FT-IR spectra (Figure 8) show the results of the samples made from different compositions with PSS (A) and PBS (B) media. The FT-IR spectra of the samples made with different hydration media are different. In every case, the spectra include two different regions. The high wavenumber part of the spectrum (3100–2800 cm^−1^) contains a contribution from C-H stretching vibrations only [44]. However, it mostly originates from the hydrocarbon chains. The low wavenumber region of the spectrum (below 1800 cm^−1^) is essentially related to the polar head groups of the lipids, as indicated. The ester ν(C=O) is usually the strongest peak near 1735 cm^−1^ and 1797 cm^−1^, followed by the phosphate contributions near 1243 cm^−1^ (ν_as_(PO_2_)) and 1103 cm^−1^(A), and 1065 cm^−1^ (B) (ν_s_(PO_2_)) [45]. The hydrocarbon chains do contribute near 1465 cm^−1^, but all-trans conformations rather absorb near 1468 cm^−1^.

The contribution of the lipid hydrocarbon chains is present in various spectral regions. The most prominent ones appear between 3050 and 2800 cm^−1^. This region essentially contains C-H stretching bands from different vibrational modes: ν_as_(CH_2_) near 2917 cm^−1^, ν_s_(CH_2_) near 2850 cm^−1^. These vibrations present several interesting features: (1) the fact that there is little overlap with other vibrations, including complex systems such as cells and tissues; (2) these vibrational modes are largely uncoupled from other modes, i.e., they do not depend on the lipid head group; and (3) they are sensitive to the structure (disordering) of the chains [44].

#### 3.4.3. Results of the Transmission Electron Microscopy (TEM) and the Atomic Force Microscopy (AFM) Spectroscopy Measurements

The liposomes are visible in the Transmission Electron Microscopy image (Figure 9A) of the 80:15:5_PPS sample. The scale in the picture is not validated for precise size determination; however, it provides a good indication of the size of the liposomes. It can be seen from the picture that the average size of the liposomes is ~100 nm. This result correlates well with the accurately defined size values obtained from the DLS measurements.

AFM measurements were also performed, giving a proper three-dimensional surface profile of the 80:15:5_PPS samples. These recordings are illustrated in the following figure (Figure 9B).

The AFM images show that liposomes with a homogeneous size distribution at nearly 100 nm were prepared. The measurement results are the same as the particle sizes obtained during the DLS and TEM measurements.

## 4. Conclusions

In conclusion, the characteristics of the desired liposomal formulation and the factors that can influence these features were defined by following the steps of the quality by design method. After performing the risk assessment, the key element of the QbD—a factorial experimental design—was developed based on the RA results. Therefore, the QbD-based product and experimental design and the liposome preparation were carried out to obtain the nanosized delivery system. Furthermore, thermosensitivity was proved, and the most stable sample with the ideal composition—the 80:15:5_PSS formula—was chosen.

## Figures and Tables

**Figure 1 molecules-27-01536-f001:**
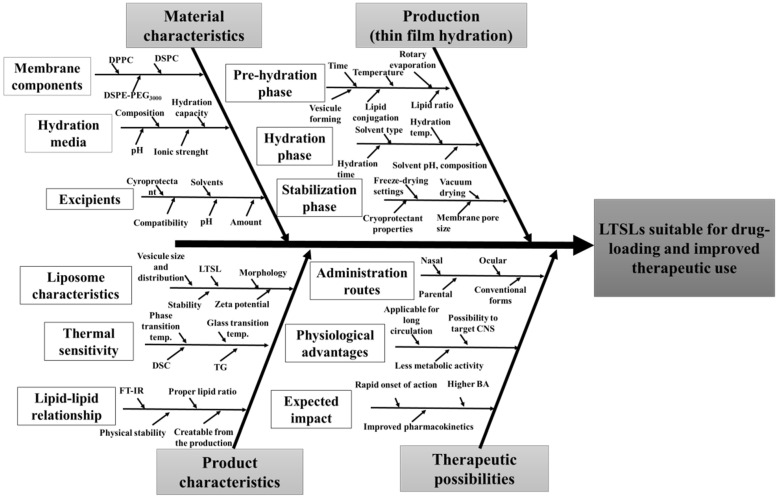
Ishikawa diagram of the target product and the related factors. Abbreviations not used before: temp., temperature; BA, bioavailability.

**Figure 2 molecules-27-01536-f002:**
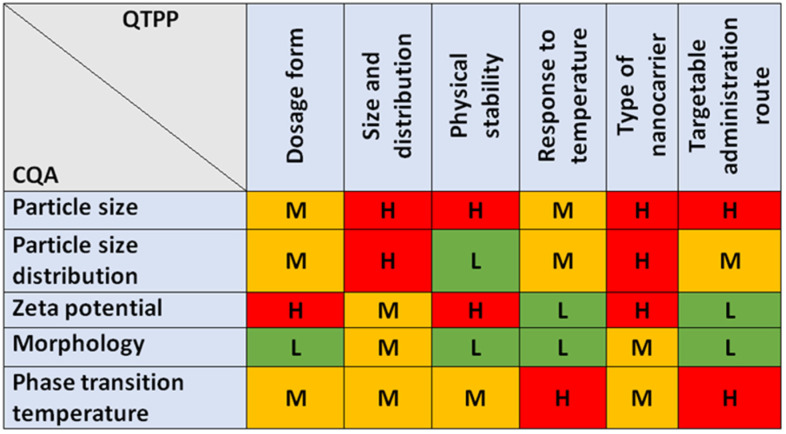
Result of the interdependence rating between the QTTPs and the CQAs.

**Figure 3 molecules-27-01536-f003:**
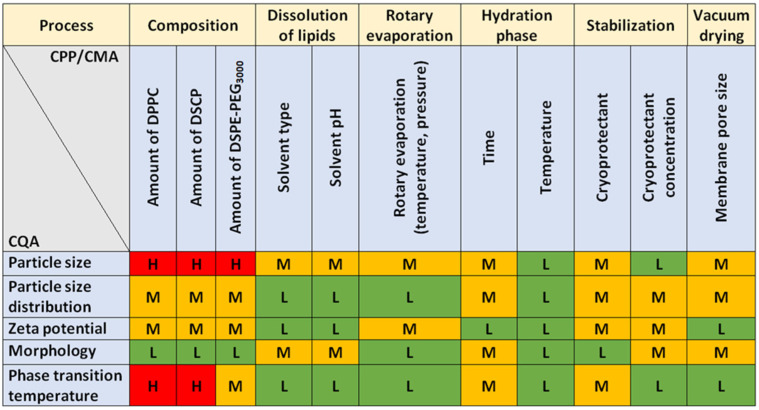
Estimation of the interrelated impacts of the critical quality (CQAs) and material attributes (CMAs), and the critical process parameters (CPPs).

**Figure 4 molecules-27-01536-f004:**
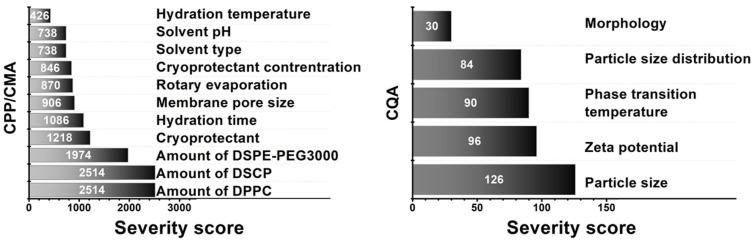
Pareto diagrams presenting the severity scores based on probability rating.

**Figure 5 molecules-27-01536-f005:**
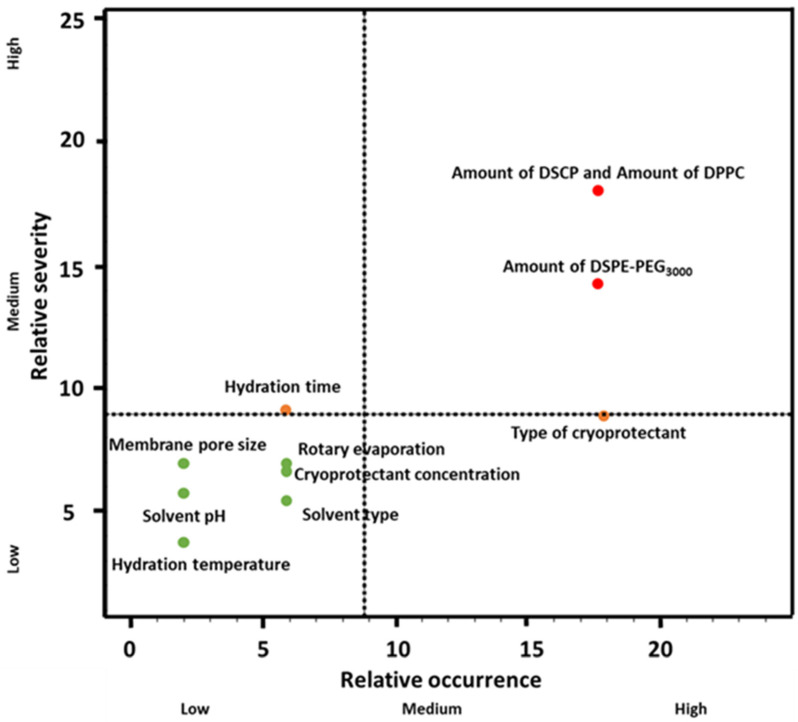
Relative severity–relative occurrence chart representing the CMA/CPP factors.

**Figure 6 molecules-27-01536-f006:**
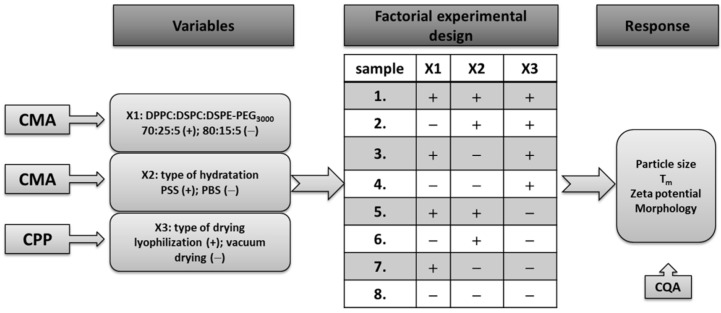
Schematic illustration of the factorial experimental design.

**Figure 7 molecules-27-01536-f007:**
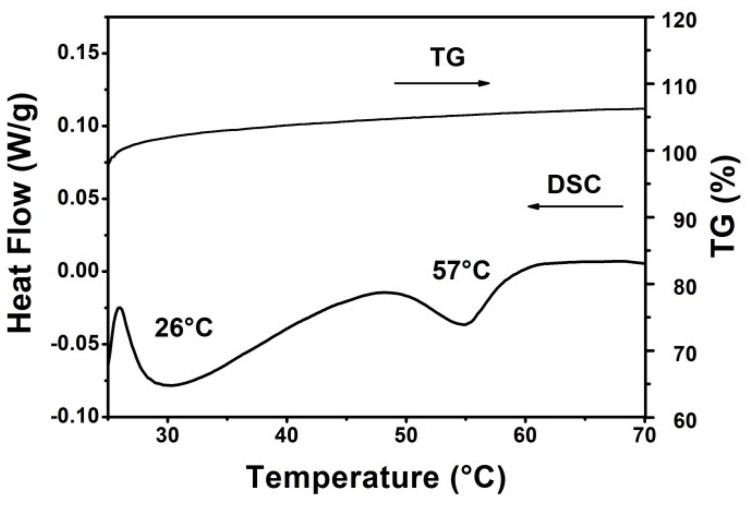
TGA and DSC curves of the 80:15:5_PSS sample.

**Figure 8 molecules-27-01536-f008:**
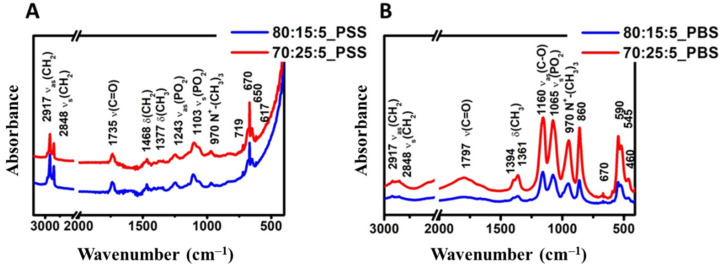
FT-IR spectra of samples made from different compositions: (**A**) Samples made with PSS medium from the compositions of 80:15:5 (blue) and 70:25:5 (red) lipid ratios. (**B**) Samples made with PBS medium from the compositions of 80:15:5 (blue) and 70:25:5 (red) lipid ratios.

**Figure 9 molecules-27-01536-f009:**
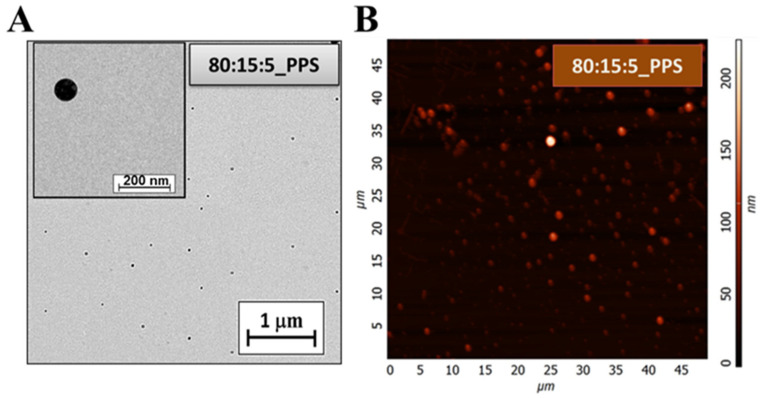
TEM (**A**) and AFM (**B**) image of the 80:15:5_PPS sample.

**Table 1 molecules-27-01536-t001:** Names and compositions of the prepared liposomes.

Name of the Samples	Mole Ratio of Lipids			Hydration Media
	DPPC	DSPC	DSPE-PEG3000	
80:15:5_PSS	80:15:5			0.9% NaCl solution (physiological saline)
80:15:5_PBS	80:15:5			pH = 7.4 PBS (phosphate-buffered saline)
70:25:5_PSS	70:25:5			0.9% NaCl solution (physiological saline)
70:25:5_PBS	70:25:5			pH = 7.4 PBS (phosphate-buffered saline)

**Table 2 molecules-27-01536-t002:** QTPP elements, their target, and the justification of LTSLs.

QTPP Element	Target	Justification
Dosage form	Liposomal colloid solution	Liquid forms can be used in multiple dosage forms—such as nasal drops, eye drops, and nasal sprays—which improve patients’ adherence to therapy.
Size and distribution	100 nm with monodisperse distribution	They can pass more efficiently across biological membranes in the appropriate nanosize range, resulting from improved dissolution and absorption profiles.
Physical stability	Stabilisable in solid form and retain the vesicle size after the dissolution	The thermodynamic stability of colloidal solutions can be achieved by the appropriate formulation, which results in stable particle size.
Response to temperature	Acquiring optimal physicochemical and particle characteristics that enable higher drug release tendencies	TLSs can be shaped to a suitable size and provide a favourable pharmacokinetic profile upon increasing temperature, which is beneficial for therapeutic use.
Type of nanocarrier	Nanosized thermosensitive liposomes	In addition to the properties of the vesicle-building materials, TLSs can increase bioavailability due to favourable physicochemical parameters.
Targetable administration route	Applicable for multiple administration routes, such as nasal, ocular, parental, etc.	Administration through alternative routes provides an opportunity to target biological compartments that conventional peroral dosage forms cannot, or if so, then poorly.

**Table 3 molecules-27-01536-t003:** Size, zeta potential, and PDI values of the samples made from different compositions measured after synthesis, two weeks later, and after heat treatment.

Sample Name	After Synthesis			Two Weeks Later			After Heat Treatment		
	Size (nm)	Zeta Potential (mV)	PDI	Size (nm)	Zeta Potential (mV)	PDI	Size (nm)	Zeta Potential (mV)	PDI
80:15:5_PSS	75 ± 2	−2.74 ± 0.67	0.25 ± 0.04	70 ± 0.5	−4.00 ± 0.48	0.24 ± 0.004	90 ± 4	−2.45 ± 0.55	0.35 ± 0.006
80:15:5_PBS	130 ± 2	−3.46 ± 1.02	0.40 ± 0.03	92 ± 0.5	−5.82 ± 2.58	0.25 ± 0.004	161 ± 1	−3.70 ± 0.23	0.39 ± 0.03
70:25:5_PSS	154 ± 5	−2.98 ± 0.93	0.34 ± 0.02	75 ± 1	−4.53 ± 1.44	0.22 ± 0.006	78 ± 1	−2.54 ± 1.21	0.22 ± 0.004
70:25:5_PBS	165 ± 1	−3.56 ± 0.59	0.34 ± 0.005	97 ± 1	−6.00 ± 2.36	0.53 ± 0.04	71 ± 1	−3.79 ± 1.19	0.360 ± 0.36

**Table 4 molecules-27-01536-t004:** Size, zeta potential, and PDI values of the 80:15:5_PSS sample at different temperatures.

T (°C)	Size (nm)	PDI	Zeta Potential (mV)
10	64 ± 1.2	0.29 ± 0.04	−2.23 ± 0.1
20	69 ± 3.2	0.24 ± 0.01	−1.42 ± 0.16
30	80 ± 1.8	0.26 ± 0.01	−1.92 ± 0.41
35	78 ± 1.1	0.26 ± 0.01	−1.54 ± 0.43
37	80 ± 0.4	0.26 ± 0.01	−1.98 ± 0.41
40	85 ± 2.8	0.27 ± 0.01	−1.39 ± 0.5

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
