# Peer review of "Pharmaceutical Development and Design of Thermosensitive Liposomes Based on the QbD Approach"

_molecules, 2022, doi:10.3390/molecules27051536_

Round 1

Reviewer 1 Report

The article “Pharmaceutical development and design of thermosensitive liposomes based on the QbD approach” by Dobo et al. has been reviewed. This work presents the preparation of thermosensitive liposomes using PEGylated lipid with the emphasis on its preparation method. Though the procedure has its merit, it is not the first report where this technique has been employed. Hence, before considering this article for publication, following concerns are to be addressed.

  1. Line 35: “Phospholipids are the main…..”. This sentence is partially correct as co-polymeric materials are equally important for liposomes development (cf. Adv. Sci.2022, 2105373; Chem. Soc. Rev., 2018,47, 8572; Biomacromolecules 2019, 20, 8, 2989). Thus, this claim requires revision.
  2. Line 284-285: “….the 4000 400 cm-1 wavelength…”. The typographical error needs correction.
  3. The red background in Figure 1 may be removed as it is difficult to concentrate on the figure.
  4. Table 4 (caption): “PdI” to be written as “PDI”.
  5. Figure 9: The red background should be removed.
  6. Figure 9B: What is the scale bar value?

Author Response

Dear Reviewer,

We submitted an original research article entitled "Pharmaceutical development and design of thermosensitive liposomes based on the QbD approach" for consideration by Molecules.

We would like to thank you and the Referees for the work in assessing our above manuscript. It is highly appreciated and welcomed. As a result, we have improved the manuscript. The text has been extended with sentences highlighting the novelty of the work, describing the used statistical methods, and improving the quality of the figures and spellings.

Corrections, changes and extensions requested by the reviewers are indicated in red in the manuscript.

Answers for Reviewer 1:

  1. Line 35: “Phospholipids are the main…..”. This sentence is partially correct as co-polymeric materials are equally important for liposomes development (cf. Adv. Sci.2022, 2105373; Chem. Soc. Rev., 2018,47, 8572; Biomacromolecules 2019, 20, 8, 2989). Thus, this claim requires revision.

You can find our answer in Line 35. “Phospholipids are important ..”

  1. Line 284-285: “….the 4000 400 cm-1 wavelength…”. The typographical error needs correction.

You can find our answer in Line 287.

  1. The red background in Figure 1 may be removed as it is difficult to concentrate on the figure.

We corrected Figure 1.

  1. Table 4 (caption): “PdI” to be written as “PDI”.

We corrected the PdI to PDI all in the manuscript.

  1. Figure 9: The red background should be removed.

We removed the red background.

  1. Figure 9B: What is the scale bar value?

We put the scalebar on the Figure 9B.

This paper has shown a Quality by Design model to develop a thermosensitive liposomal formulation, first in the literature. We also aimed to check the general effects of some critical factors via a risk assessment-based approach in the development process of the liposomes, which is significant in designing these products.

The kind work and the valuable advice of the Reviewers are highly appreciated.

Sincerely,

Dorina Gabriella Dobó

Reviewer 2 Report

This article is of high quality and forms the basis for very cardinal future work. Overall, the scientific basis is excellent and the authors describe their intentions early and follow through. There are a few concerns that I highlight below

  • The article needs substantive English editing as in some places it appears the words chosen don't quite drive home the point.
  • The materials and methods section needs a lot of work. In many parts the authors do not give details about the manufacturers of equipment. Some of the methods are not described in enough detail to replicate them (i.e PBS)
  • Table 1 appears rather abruptly with very little information as to what is coming up
  • The images should be enhanced and not include a red background for the Ishikawa diagram and the Microscopy images.
  • On Page 12, It would be very cardinal to include the references in the statement "according to literature..."
  • The second paragraph on page 14 needs further work and explanation as it does not adequately explain the phenomenon being evidenced.
  • Figure 9, it would be best to use a TEM micrograph with more than one liposome in the image

The data produced in this paper are very important in the potential treatment of liposomes by thermo-triggered nanomedicines. However, there needs to be more reference and scientific basis for considering the technology to be thermosensitive in payload release other than an in increase particle size. As these were made as a proof of concept, it would be ideal to attempt an encapsulation of a proof of concept API

Author Response

Dear Reviewer, 

We submitted an original research article entitled "Pharmaceutical development and design of thermosensitive liposomes based on the QbD approach" for consideration by Molecules.

We would like to thank you and the Referees for the work in assessing our above manuscript. It is highly appreciated and welcomed. As a result, we have improved the manuscript. The text has been extended with sentences highlighting the novelty of the work, describing the used statistical methods, and improving the quality of the figures and spellings.

Corrections, changes and extensions requested by the reviewers are indicated in red in the manuscript.

  1. The materials and methods section needs a lot of work. In many parts, the authors do not give details about the manufacturers of equipment. Some of the methods are not described in enough detail to replicate them (i.e PBS)

The experimental part was described in detail, supplemented where necessary.

  1. Table 1 appears rather abruptly with very little information as to what is coming up.

Table 1. has been moved to page 6 and we have supplemented the relevant part.

  1. The images should be enhanced and not include a red background for the Ishikawa diagram and the Microscopy images.

We corrected the images.

  1. On Page 12, It would be very cardinal to include the references in the statement "according to literature..."

You can find our answer in Line 420.

  1. The second paragraph on page 14 needs further work and explanation as it does not adequately explain the phenomenon being evidenced.
  2. Figure 9, it would be best to use a TEM micrograph with more than one liposome in the image

DSC and TEM measurements were repeated several times. The new results were expanded and included in the manuscript, highlighted in red.

  1. The data produced in this paper are very important in the potential treatment of liposomes by thermo-triggered nanomedicines. However, there needs to be more reference and scientific basis for considering the technology to be thermosensitive in payload release other than an in increase particle size. As these were made as a proof of concept, it would be ideal to attempt an encapsulation of a proof of concept API

You can find our answer in Line 166. “None of the formulations contained active pharmaceutical ingredients (API)”. Future work will include API, we would like to write more about API another manuscript.

This paper has shown a Quality by Design model to develop a thermosensitive liposomal formulation, first in the literature. We also aimed to check the general effects of some critical factors via a risk assessment-based approach in the development process of the liposomes, which is significant in designing these products.

The kind work and the valuable advice of the Reviewers are highly appreciated.

Sincerely,

Dorina Gabriella Dobó

Round 2

Reviewer 2 Report

The article forms extremely important basis for future work. It is well thought out and flows well.